# Evaluation of Pollution Level, Spatial Distribution, and Ecological Effects of Antimony in Soils of Mining Areas: A Review

**DOI:** 10.3390/ijerph20010242

**Published:** 2022-12-23

**Authors:** Shuting Zhao, Taoran Shi, Akihiko Terada, Shohei Riya

**Affiliations:** 1Department of Chemical Engineering, Tokyo University of Agriculture and Technology, Tokyo 184-8588, Japan; 2School of Applied Meteorology, Nanjing University of Information Science & Technology, Nanjing 210044, China

**Keywords:** antimony, soil, spatial distribution, chemical species, microbial community, health risk

## Abstract

The first global-scale assessment of Sb contamination in soil that was related to mining/smelting activities was conducted based on 91 articles that were published between 1989 and 2021. The geographical variation, the pollution level, the speciation, the influencing factors, and the environmental effects of Sb that were associated with mining/smelting-affected soils were analyzed. The high Sb values mainly occurred in developed (Poland, Italy, Spain, Portugal, New Zealand, Australia) and developing (China, Algeria, Slovakia) countries. Sb concentrations of polluted soil from mining areas that were reported in most countries significantly exceeded the maximum permissible limit that is recommended by WHO, except in Turkey and Macedonia. The soil Sb concentrations decreased in the order of Oceania (29,151 mg/kg) > North Africa (13,022 mg/kg) > Asia (1527 mg/kg) > Europe (858 mg/kg) > South America (37.4 mg/kg). The existing extraction methods for Sb speciation have been classified according to the extractant, however, further research is needed in the standardization of these extraction methods. Modern analytical and characterization technologies, e.g., X-ray absorption spectroscopy, are effective at characterizing chemical speciation. Conditional inference tree (CIT) analysis has shown that the clay content was the major factor that influenced the soil Sb concentration. Non-carcinogenic risks to the public from soil Sb pollution were within the acceptable levels in most regions. An Sb smelter site at the Endeavour Inlet in New Zealand, an abandoned open-pit Sb mine in Djebel Hamimat, Algeria, an old Sb-mining area in Tuscany, Italy, and Hillgrove mine in Australia were selected as the priority control areas. *Cynodon dactylon*, *Boehmeria*, *Pteris vittata*, and *Amaranthus paniculatus* were found to be potential Sb accumulators. All of the values of bioaccumulation factors for the crops were less than one. However, ingestion of Sb through crop consumption posed potential non-carcinogenic health risks, which should not be neglected. The soil variables (pH, Eh, total sulfur, carbon nitrogen ratio, total organic carbon, and sulfate), the total Sb and the bioavailable Sb, and heavy metal(loid)s (As, Pb, and Fe) were the major parameters affecting the microbial community compositions.

## 1. Introduction

Antimony (Sb) is non-essential for living organisms and is an emerging toxic metalloid. Sb is an important metalloid that is widely used in flame retardants, battery and Pb alloys, glass ceramics, chemical products, catalysts, and in military affairs (Filella et al., 2002) [1]. Sb and its compounds have been classified as pollutants of priority interest and as hazardous substances by the United States Environmental Protection Agency (US EPA), the European Union (EU), and the World Health Organization (WHO) (Filella et al., 2002) [1]. The Ministry of Ecology and Environment of the People’s Republic of China released “Opinions on further strengthening the prevention and control of heavy metal pollution” on 7 March 2022 and proposed that Priority should be given to prevention and control of Sb in China. (https://www.mee.gov.cn/xxgk2018/xxgk/xxgk03/202203/t20220315_971552.html, accessed on 7 March 2022). Sb can be transported over long distances, causing harm to the environment and to human beings, not only where they are produced and consumed, but also on a global scale (Daellenbach et al., 2020) [2]. The evidence of moss species has demonstrated that Sb has been transported to the remotest parts of northern Norway (Berg and Steinnes, 1997) [3]. The research concerning ice cores from Mt. Qomolangma has also shown the long-term historical record of Sb atmospheric transport and deposition (Hong et al., 2009) [4]. There has been a 50- to 100-fold maximum increase in Sb in the air during the 20th century compared with back-ground value (Cloy et al., 2005) [5]. Heavy metals may adsorb into aeolian particles, and they can be transported over long distances (Grousset et al., 1995) [6]. Human and animal bodies can be exposed to Sb in the environmental media (air, water, soil, and plants) through contact with the skin, via inhalation, and through the food chain. Sb has a strong affinity for sulphydryl (-SH) in proteins (Gebel, 1997) [7]. The combination of Sb and -SH in the human body can inhibit the activity of mercapto-iodoacetamide, can interfere with the metabolism of proteins and sugars, can damage the liver, the heart, and the nervous system, and can stimulate the mucosa (Agency for Toxic Substances and Disease Registry, Atlanta, 2019) [8]. Occupational exposure and the intake of foods/drugs are the main causes for acute intoxication. It has been estimated that the range of occupational cancer risk is 0.025 (25 in 1000) to 0.11 (110 in 1000). However, the limit of the cancer risk for occupational inhalation exposure of Sb trioxide has been estimated to be 0.096 (96 in 1000) (Schildroth et al., 2020) [9]. Additionally, dietary exposure to Sb is far lower than that of occupational exposure (Belzile et al., 2011) [10]. Natural processes (e.g., volcanic activity and rock weathering) and anthropogenic activities (e.g., mining, beneficiation, smelting, Sb consumption, brake wear, and coal combustion) can release Sb into the atmosphere (He et al., 2019; Zhu et al., 2020) [11,12]. According to estimates, about (4.7–47) × 10^6^ kg of Sb is released into the soil each year, based on a quantitative evaluation of heavy metal contamination in soils (Li et al., 2011) [13].

Mining is the largest anthropogenic source of Sb pollution into the soil environment (He et al., 2019; Li et al., 2018; National pollutant inventory, 2019–2020 data within Australia) [11,14,15]. Sb accumulation in the environment can be substantially accelerated by mining and smelting. China has the largest Sb reserves in the world. The high levels of Sb in the soil on the one hand reflect an Sb-rich natural background (MEMSC, 1990) [16]. The dust particles originated from mining/smelting processes and open tailing ponds. Atmospheric precipitation or direct contact with tailings and wastewater may easily result in soil and water pollution. The acceptance of Sb mine exploration and mining permit application was suspended in order to control the total quantity of Sb exploitation for the first time in China in 2009 (http://mnr.gov.cn/fw/zwdt/gsgg/200905/t20090506_2084936.html, accessed on 6 May 2009). The EU listed Sb as a critical raw material in 2010 (European Commission, 2010) [17]. The soils in the Sb mining areas have shown severe Sb contamination in many reported studies (Qin et al., 2022) [18]. Thus, there is an urgent need to generally review the soil environmental Sb pollution that is associated with mining and smelting. As a prerequisite to environmental contamination remediation and management, and the implementation of reduction programs, baseline information is required.

There are still no uniform international standards of soil Sb. For instance, Sb toxicity guidelines in the agricultural/residential parkland and the commercial/industrial soils of Canada are 20 and 40 mg/kg, respectively (CCME, 2018) [19]. The US EPA has set ecological screening levels of soil mammals (0.27 mg/kg) and invertebrates (78 mg/kg) (USEPA, 2005) [20]. France has set soil source definition values for sensitive (100 mg/kg) and non-sensitive (250 mg/kg) areas (Carlon, 2007) [21]. The WHO has set 36 mg/kg as the safe soil concentration for Sb, based on human health risks (Chang et al., 2002) [22]. The screening values of Sb are 20 (Class I land) and 180 (Class Ⅱ land) mg/kg and the control values are 40 (Class I land) and 360 (Class Ⅱ land) mg/kg, based on the soil environmental quality–risk control standard for soil contamination of development land (GB 36600-2018) in China. At present, although there is no unified international standard for Sb content in soil, it is urgent to prevent and control Sb pollution in soil effectively. The establishment of a set of effective evaluation systems for soil Sb pollution is an important indicator for soil Sb pollution control and can also help to remediate contaminated soil in a timely manner.

Previous review articles have covered a wide field, such as the pollution sources, the distribution and speciation in different environment mediums, the biogeochemical processes, the toxicity, and the ecological effects (Amarasiriwardena and Wu, 2011; Shotyk et al., 2005) [23,24]. A majority of the concerns during recent decades have been in regards to the geochemistry and the behavior of Sb in environmental media (Herath et al., 2017; Wilson et al., 2010) [25,26]. Very recently, a review of Sb contamination, the consequences, and the removal techniques was presented by Li et al. (2018) [27]. He et al. (2019) and Pierart et al. (2015) [11,28] reviewed the Sb speciation and bioavailability in the environment, which improved our understanding of the biogeochemical processes and the ecological effects. Chu et al. (2019) [29] analyzed the Sb flow in different spheres of the earth in China. To date, the global Sb spatial distribution, the pollution level, and its ecological effects that are associated with mining/smelting activities are not fully understood. This study summarizes the published data on Sb concentrations in mining/smelting-affected soils and assesses their potential environmental effects on human health, plants, and microorganisms. This study provides a scientific basis for assessing the environmental quality in Sb mining areas and establishes an effective soil Sb pollution early warning system.

## 2. Data Sources and Collection

The data were extracted from Elsevier Science Direct, Science Online, and Web of Science. In total, 91 papers that were published between 1989 and 2021 were collected using the keywords Antimony or Sb, soil, and mining. For the literature screening, the following filter criteria were applied to these papers (Appendix A, Appendix A). The studies that were unrelated to mining were excluded (Benhamdi et al., 2014; Hammel et al., 1998; Jurkovič et al., 2019; Lewińska et al., 2016; Lewińska et al., 2018; Villarroel et al., 2006; Yang and He, 2015) [30,31,32,33,34,35,36]. The studies in which soil Sb concentrations were not clearly defined (Wei et al., 2011) [37], including review articles, were also excluded. The data on the Sb concentrations in the soil were collected and processed from the screened literature, and the standard measurement unit of soil Sb concentration was mg/kg. The sampling and processing methods that were used in these studies were all compliant with the standards and guidelines. Across the studies, hydride generation-atomic fluorescence spectrometry was the most commonly applied method for measuring the soil Sb concentrations.

Data on soil Sb concentrations from 72 sampling sites in mining/smelting regions were obtained. The geographical locations of the studies that were related to Sb mining/smelting, which were distributed across 5 continents and 21 countries, including a few European countries, Bolivia, China, Japan, Iran, Australia, New Zealand, and Algeria, were retrieved from the selected studies (Figure 1). We found several relevant studies on Sb pollution in the soil caused by mineral development in European countries and China (Figure 1), indicating that Sb soil pollution that was caused by mining activities has gained wide attention in these regions. Each of these studies examined only one or several regions that were suspected of pollution. For example, the Xikuangshan Sb mine has attracted much interest from scientists around the world (He 2007; Li et al., 2014; Okkenhaug et al., 2011; Wang et al., 2018) [38,39,40,41].

## 3. Results and Discussion

### 3.1. Geographical Variation and Pollution Levels

A total of 552 measurements of Sb concentrations in soils were obtained from the literature. The geographical variations in the Sb concentrations in the soils of different regions were investigated based on the existing data (Figure 2). The results showed that 72% of the soils had Sb concentrations that were above the maximum permissible limit (36 mg/kg) that is recommended by WHO. Some specific regions from China and the European countries are enlarged in Figure 2. The Sb deposits are scattered across the world; however, they differ significantly in terms of their abundance. Based on the global distribution of Sb resources (Appendix A) (Labay et al., 2017) [42], China is the richest in Sb resources, followed by Russia, Bolivia, Kazakhstan, South Africa, and Turkey. Sb mine production in 2021 was the highest in China (60,000), followed by Russia (25,000), Tajikistan (13,000), Burma (2000), Bolivia (2700), Turkey (1300), Australia (3400), and Vietnam (400) (in metric tons) (https://www.chyxx.com/industry/1115956.html accessed on 8 July 2022). Considering the levels of economic development, the scientific and technical levels, the emission limits of air pollutants for typical industries, and the field emission test results, the countries were divided into five regions (Appendix A) (Zhu et al., 2020) [12]. Regions one and two represent developed countries, while regions three to five represent developing countries and districts.

Based on the existing data, Sb was mainly found in the soils of regions one, two, three, and four (Appendix A). It is interesting that the high Sb values mainly occurred in regions one and two, i.e., the developed countries (Appendix A). After calculating the average value of each paper, the soil Sb concentration was up to 5650 mg/kg in the historical mining sites in Radzimowice, Olszanka, and Poland (Lewińska and Karczewska, 2019) [43]. In Italy, the average value of soil Sb concentration in an old Sb mining area reached as high as 5598 mg/kg in Tuscany (Baroni et al., 2000) [44] and 1338 mg/kg at the Su suergiu abandoned mine in Sardinia (Cidu et al., 2014) [45]. In Spain, the peak values occurred in the quartz–stibnite vein deposits in an abandoned mining area in the towns of Losacio and Cogollas, Zamora Province (1782 mg/kg) (Casado et al., 2007) [46] and the Sb mining areas in Extremadura (1324 mg/kg) (Murciego et al., 2007) [47]. In Portugal, high soil Sb concentrations were detected in the Sarzedas mine in the Castelo Branco County (663.1 mg/kg) (Pratas et al., 2005) [48] and in the São Domingos copper sulfide mine in the Baixo Alentejo Province (467.8 mg/kg) (Anawar et al., 2013) [49]. The soil Sb concentrations in New Zealand and Australia (region two) were very high. The soil Sb concentration in a historic Sb smelter site at the Endeavour Inlet in New Zealand was 80,000 mg/kg (Wilson et al., 2004) [50]. The soil Sb content in the Hillgrove mine in Australia was 3626 mg/kg (Wilson et al., 2013) [51]. Some of the sites in regions three and four also had high soil Sb values. The soil Sb concentration in an abandoned open-sky Sb mine in Djebel Hamimat, Algeria, in North Africa was 15,549 mg/kg (Benhamdi et al., 2014) [30]. The soil Sb concentrations in the Banpo Sb mine in Guizhou Province (1993 mg/kg) (Ning et al., 2015) [52], in the Xikuangshan Sb mine in Hunan Province (1772 mg/kg) (He 2007) [38], and in the Guibei-Qiannan Sb ore concentration area (1527 mg/kg) (Li et al., 2020) [53] in China were extremely high. The soil Sb concentrations in the abandoned Sb mines in Poproč and Dúbrava in Slovakia were 1612 and 1474 mg/kg, respectively (Hiller et al., 2012) [54]. In mining process, anthropogenic factors such as ore mining, ore stockpiling and transport, and mining wastewater discharge are the important factors in causing Sb pollution in mining areas and adjacent regions.

The soil Sb concentrations that are related to mining activities among the countries from different continents in the world were compared (Figure 3). The soil Sb concentrations in most of the countries substantially exceeded the threshold value (36 mg/kg), except for Turkey and Macedonia. The soil Sb concentrations ranged from 3626 mg/kg in Australia to 80,200 mg/kg in New Zealand in Oceania. The soil Sb concentration was 13,021 mg/kg in Algeria in North Africa. In Europe, the soil Sb concentrations ranged from 17.5 mg/kg in Macedonia to 1378 mg/kg in Slovakia. In Asia, the soil Sb concentrations ranged from 8.53 mg/kg in Turkey to 2542 mg/kg in Japan. The soil Sb concentration in Bolivia in South America was 37.4 mg/kg, which is marginally above the WHO guideline for soil Sb (36 mg/kg). The soil Sb concentrations were the highest in Oceania, followed by North Africa, Europe, Asia, and South America. The distribution of the study areas varied between the continents, which could have biased the results.

### 3.2. Chemical Speciation and Bioavailability

The Sb speciation in the soil that was related to mining activity, which is based on existing studies, has been summarized in Table 1. One limitation is that the soluble fractions were obtained through numerous different extraction protocols, leading sometimes to potentially misleading and incomparable results. The common sequential extraction procedures are BCR (Carvalho et al., 2012; Huang et al., 2019; Protano and Nannoni, 2018) [55,56,57] and the Tessier method (Deng et al., 2020; He 2007; Ning et al., 2015) [38,52,58]. The most frequently used single extraction reagents are H_2_O (Casado et al., 2007; Ettler et al., 2007; Flynn et al., 2003; Li et al., 2020; Murciego et al., 2007; Okkenhaug et al., 2011; Pérez-Sirvent et al., 2012; Vaculík et al., 2013; Wei et al., 2015) [40,46,47,53,59,60,61,62,63], acetic acid (Baroni et al., 2000; Vaculík et al., 2013) [44,62], HCl (Tan et al., 2018) [64], tartaric acid (Tan et al., 2018) [64], citric acid (Tan et al., 2018) [64], NH_4_NO_3_ (Ettler et al., 2007; Gál et al., 2006; Lewińska et al., 2018; Vaculík et al., 2013) [43,59,62,65], CaCl_2_ (Ettler et al., 2007; Lewińska et al., 2018; Tan et al., 2018) [43,59,64], Na_2_HPO_4_ (Ettler et al., 2007; Tan et al., 2018) [59,64], EDTA (Mariet et al., 2016; Tan et al., 2018; Vaculík et al., 2013) [62,64,66], and DTPA (Ettler et al., 2007) [59]. By comparing single extraction and sequential extraction, citric acid and tartaric acid showed higher extractability for Sb (Tan et al., 2018) [64]. However, the extracted Sb amounts that were obtained from the same extractant were different for each study. For example, the extraction efficiencies with NH_4_NO_3_ (1 M) for Sb were 0.86% for the forest soils and 0.89% for the tilled soils (Ettler et al., 2007) [59], 0.02–0.66% (Lewińska et al., 2018) [43], and 0.004–1.13% (Vaculík et al., 2013) [62]. Diffusive gradients in thin films (DGT) were used in order to investigate the bioavailable Sb in the soil. The concentration of Sb that was measured by DGT was 11.55–876.6 μg/L in the soil from the Xikuangshan Sb mine, China (Wang et al., 2018) [41]. The contrastive research on DGT and sequential extractions has shown that those two methods predicted the acclamation of Sb in radish (*Raphanus sativus*) well. The regression coefficients between the Sb in shoots and soil were 0.98 for the DGT method and were 0.96 for the sequential extraction procedure (Ngo et al., 2016) [67]. As the unit of bioavailable Sb concentration that is measured by DGT is not compatible with that from other methods, a comparison with other methods was not possible. The comparison of Sb chemical species in the soils of different regions is difficult because of the differences in the extraction methods, the reagents, and/or the operating conditions that have been used for chemical speciation analysis. Though the same chemical extractants are applied, the reaction conditions (e.g., the duration of oscillation, the water to soil ratio, and the extractant concentration) may be different. Currently, there is a lack of unified extraction methods. Sb is present primarily in the residual fraction in soils, indicating low Sb bioavailability (Carvalho et al., 2012; Deng et al., 2020; He 2007; Ning et al., 2015; Protano and Nannoni, 2018) [38,52,55,57,58]. Studies on the bioaccessibility of Sb in mining/smelting-affected soils are very limited. The average bioaccessibility values of Sb in soils from the Xikuangshan Sb mine, Hunan, China, was 5.89 ± 6.44% for the simplified bioavailability extraction test (SBET), 7.83 ± 9.82% for the gastric phase, and 3.03 ± 3.53% for the intestinal phase (Li et al., 2014) [39]. The Sb chemical valence in the soils from the Xikuangshan Sb mine was dominated by Sb(V) (0.59–10.15%) and the highest ratio for Sb(III) was only 0.04% (Okkenhaug et al., 2011) [40]. The speciation analysis of Sb is the most commonly used chemical and destructive extraction method. However, the extractant selectivity lacks specificity and can only provide speciation in an operational sense. This may lead to some deviations in the understanding of the speciation and the bioavailability of Sb in comparison with its initial speciation in the environment. X-ray absorption spectroscopy is an effective technical method that can be used in order to characterize elemental speciation. It is a really valuable tool for the samples that are more reactive in order to produce redox reaction during sample preparation. Extended X-ray absorption fine structure (EXAFS) analysis can measure the valence state and structure of Sb, showing that Sb(III) is bound to three oxygen atoms and Sb(V) combines with six oxygen atoms. This technique can also obtain the main host of Sb in a water environment. X-ray absorption fine structure (XAFS) analysis has revealed that Sb mostly occurred as Sb(V) in the form of FeSbO_4_ in the soils (Park et al., 2021) [68]. X-ray absorption near-edge structure (XANES) analysis also indicated that Sb(V) was the dominant form in the soil samples from the Xikuangshan Sb mine (Okkenhaug et al., 2011) [40]. It was also reported that the soil Sb in Japan was mainly in the state of Sb(V), based on XANES analysis (Mitsunobuet al., 2006) [69]. However, the main disadvantage of this technique is low sensitivity. This technique only applies to the environmental samples (collected from near to the Sb mining area, smelter, or shooting range) with a high Sb concentration.

### 3.3. Influencing Factors of Soil Sb Accumulation

The data of the soil physicochemical properties and the corresponding Sb concentrations were extracted from the existing data. Only the effects of soil physicochemical properties that were backed by the data on the Sb accumulation in the soil were assessed (Figure 4). By constructing conditional inference tree (CIT) models, this study estimated the relationship between the Sb content and the predictor factors. Detailed information on the CIT model is described in the Appendix A.

It was found that the clay content and the electrical conductivity (EC) were positively correlated with the Sb concentrations in soil (*p* < 0.05) (Figure 4). It has been reported that the EC was positively associated with the Sb(III) percentage of the sediments (Liang et al., 2018) [71]. The EC could be considered as a reflection of the intensity of Sb species transformations. There was a significantly negative correlation between the pH and the EC (*p* < 0.01) (Figure 4). Soil acidification is accompanied by an increase in the ion concentrations in the soil, which causes an increase in the soil conductivity. The clay content was the major factor that influenced the soil Sb concentration, as was observed through the CIT analysis (Figure 5). The main components of clay are phyllosilicate and iron oxide minerals. The Sb level in soils is positively correlated with the silicate minerals and the quartz contents (García-Lorenzo et al., 2015) [72]. Adsorption is one of the most important reaction mechanisms of Sb in soil. The clay minerals play a critical role in retaining Sb by complexation processes, which can limit the transport and the bioavailability of Sb greatly (Zhang et al., 2022) [73]. Anion adsorption in clay minerals is related to broken edges of clay particles, which typically occur by surface ligand exchange (Wilson et al., 2010) [26]. McBride (1994) [74] reported that clay minerals are potentially effective geological sorbents and sinks for soluble Sb. The maximal sorption capacities of bentonite for Sb(III) and Sb(V) were 370–556 and 270–500 μg/g, respectively (Xi et al., 2011) [75]. Moreover, owing to the high surface area to volume ratios, the clay particles possess a high binding capacity for metal(loid) ions and are attractive microhabitats for microorganisms (Duester et al., 2007) [76].

Besides, there are other factors, which are not shown in Figure 5, that also influence Sb accumulation in soil. In the moderate reductive soil, Sb binds to relatively unstable Fe/Al hydrous oxide. The photooxidation of Sb(III) occurred through electron transfer from Sb(III) to Fe(III) along with the reduction of Fe(III) to Fe(II) through a ligand-to-metal charge-transfer process. Sb(V) can be reduced to Sb (III) in soils in the presence of Fe^2+^ (Fan et al. 2017) [77]. In soils with a high organic matter content, Sb readily binds to soil organic colloid. Tserenpil et al. (2011) [78], pointed out that the adsorption of Sb(III) by humic acid can reach 50% under low pollution conditions. Sb (III) oxidation was controlled by MnO_2_, and Sb (V) sorption occurred at the edge sites of MnO_2_ (Sun at al., 2018) [79]. Similarly to MnO_2_ oxide, iron has ability to oxidize Sb (III).

### 3.4. Toxicity and Ecological Effects of Sb

#### 3.4.1. Health Risk of Exposure to Soil Sb

The carcinogenic slope factors of Sb are not all available; only the non-carcinogenic risk of Sb was estimated in this study. The non-carcinogenic risk was assessed according to the models that have been produced by the US EPA. According to the US EPA (2000) [80], if the hazard index (HI) is less than one, the exposed individual is assumed to be safe. In contrast, if the HI exceeds one, non-carcinogenic effects may occur. The combination of non-carcinogenic risk in humans from the different exposure pathways can be estimated by adding the HI of each exposure pathway, which in this case included ingestion, dermal contact, and inhalation. The health risk assessment method and the exposure factors for models are described in detail in the Appendix A.

The cumulative probabilities of the HI of adults and children were assessed based on the results of the non-carcinogenic risk assessment (Figure 6). The non-carcinogenic risks in most of the areas were at acceptable levels (Figure 6), demonstrating a relatively low non-carcinogenic risk to the public in most of the regions. The HI for two populations varied greatly, in the order of adults > children, with no statistically significant differences. The main reason for this is the accumulation of Sb in the human body. For adults, the non-carcinogenic risks from soil Sb exposure in approximately 9.57% of the sampling sites were higher than one. For children, 8.29% of the areas had HI that was greater than one. Although the non-carcinogenic risks of Sb exposure to humans through soil in most parts of the regions were within the acceptable level, some regions are still greatly worthy of our attention. The HI value of adults in the abandoned open-sky Sb mine on the slope of Djebel Hamimat mountain in Algeria was 21.4 (Benhamdi et al., 2014) [30]. The non-carcinogenic risk values of adults in the historic Sb smelter site in New Zealand were approximately 26 times higher than the acceptable levels (Wilson et al., 2004) [50]. Thus, the non-carcinogenic risks in these areas deserve more attention, and regular monitoring of Sb in the soils is strongly recommended. Traditional human health risk assessments are based on the total contaminant intake, however, only the pollutants that are soluble in human digestive juices (i.e., the bioaccessible fraction) can be absorbed into the human body. Studies have shown that Sb in soil is mainly in the form of a relatively stable residue, which is difficult to be extracted by human digestive juices (Bolan et al., 2022) [81]. Compared with the total amount of Sb, an assessment based on the bioavailability of Sb in the soil can obtain significantly lower exposure and human health risk levels (Denys et al., 2009) [82]. It can be seen that the traditional assessment methods might overestimate the risks, leading to excessive control measures. Therefore, accurately assessing the human health risks that are caused by exposure to contaminated soil is of great significance for the scientific control and the remediation of contaminated soil.

#### 3.4.2. Plants/Crops Uptake

The ability of plants/crops to accumulate Sb can be estimated using the bioaccumulation factor (BCF). The BCF is the ratio between the concentration of Sb in the root/shoot of a plant and that present in the soil. As shown in Figure 7, the BCF of *Boehmeria*, *Equisetum arvense* L., and *Pteris Vittata* in the aboveground parts for Sb were 2.17, 1.58, and 1.21, respectively. In the aboveground parts of *Boehmeria*, *Equisetum arvense L*., and *Pteris vittata*, the concentration of Sb was 477.53–745.76 (Xue et al., 2014) [83], 8.2 (Zhang et al., 2009) [84], and 321.5 mg/kg (Wan et al., 2017) [85], respectively. *Boehmeria,* as one kind of local major optimal plant, is characterized by a strong adaptation ability, fast-growing capabilities, and a high biomass. It grows quickly and shapes the local small community in a high Sb pollution environment. It has been shown that *Pteris vittata* accumulates more Sb(III) than Sb(V) (Tisarum et al., 2014) [86]. The BCF of *Cynodon dactylon* and *Amaranthus paniculatus* in the below-ground parts for Sb were 4.2 and 2.5, respectively. *Cynodon dactylon* exhibited a strong ability to concentrate Sb in the roots (502.87–2180.34 mg/kg) (Xue et al., 2014) [85]. The roots of *Amaranthus paniculatus* accumulated Sb up to a maximum of 716.16 mg/kg (Xue et al., 2014) [85]. The accumulation of Sb in the below-ground parts of *Dittrichia viscosa*, *Plantago lanceolata*, and *Sedum lineare Thunb* reached up to 1136 (Murciego et al., 2007) [47], 1150.3 (Baroni et al., 2000) [44], and 21,148 mg/kg (Ning et al., 2015) [52], respectively. However, the BCF of these species were less than one or were not available due the lack of information on soil Sb content. So far, Sb hyperaccumulators have yet to be found, and the threshold levels of Sb in hyperaccumulators have not been reported. In addition, there are a few plants with Sb tolerance, such as *Pteris cretica* (Feng et al., 2011) [87] and *Brassica Juncea* (Barajas-Aceves et al., 2012) [88], and some plants with Sb enrichment ability, such as *Miscanthus sinensis* (Xue et al., 2015) [89] have also been discovered. *Miscanthus sinensis* is similar to an energy plant and it is one of the most ideal repair materials for Sb pollution. Further research is needed in order to validate whether these are potential Sb hyperaccumulator species. In practice, the soils in mining areas are often contaminated by multiple heavy metals, and the identified hyperaccumulators that only tolerate and accumulate Sb are inadequate for phytoremediation. Screening appropriate multi-metal hyperaccumulators is key to the success of phytoextraction for the contaminated soils in mining sites.

Based on previous studies, all of the BCF values for the crops that have been referred to here were low (<1), even in the presence of high concentrations of Sb in the soils (He 2007) [38] (Figure 8). The BCF values of brown rice (0.24) and radish (0.14) in the aerial parts for Sb were comparatively high. The BCF values of rice (0.29) and *Arachis hypogaea* L. (0.17) in the underground parts for Sb were also comparatively high (Figure 8). It was found that the BCF values of the edible crop tissues of brown rice and *Arachis hypogaea* L. were high, which may have adverse impacts on human health. It is worth noting that a low BCF value does not necessarily imply low concentrations in the crop. *Arachis hypogaea* L., *Daucus carota* L., and *Allium sativum* L. showed good bioconcentration ability towards Sb, with levels of 314 (Okkenhaug et al., 2011) [40], 23.72 (Zeng et al., 2015) [90], and 25.37 mg/kg (Zeng et al., 2015) [90], respectively, in the below-ground parts (the edible parts). The accumulation of Sb in vegetables and crops can be directly or indirectly ingested by humans through the food chain and subsequently accumulate in human body cells, which may raise health risks (Fan et al., 2017; Pierart et al., 2015) [28,77]. Wu et al. (2011) [91] showed that dietary exposure was the predominant source of Sb exposure in residents in the Sb mining area. The results of the health risk assessment showed that the consumption of vegetables around the Xikuangshan mine may increase the health risks of the residents (Zeng et al., 2015) [90]. The health risks of the Sb in the vegetables to the local inhabitants were much higher than that of As (Zeng et al., 2015) [90]. 

Sb intake through brown rice consumption posed potential non-carcinogenic risks to the local populations of the Qibaoshan mine, the Shuikoushan mine, and Stannary in Hunan, China, (Fan et al., 2017) [77]. These previous studies calculated the HQ values of Sb from the crop intake based on the assumption that all of the Sb that was contained in crops was absorbed by human beings, which overestimated the health risk. Only 5–20% of the total Sb content that is ingested can be absorbed through the digestive tract (Pierart et al., 2015) [28]. It is necessary to investigate the speciation, the bioaccessibility, and the bioavailability of Sb in the crops that are consumed by the inhabitants of the regions surrounding the mining areas. In the mining areas, the estimated average daily intake of heavy metals from vegetables was correlated with the heavy metal contents in the soil and the water. Thus, it is necessary to establish a soil ecological screening value and a toxicity threshold of Sb for crops in order to ensure that the accumulation of Sb in the edible parts of crops is at a safe level.

#### 3.4.3. Microbial Characteristics

##### Microbial Diversity and Population

The bacterial communities in soils originating from Sb mining areas have been awarded more attention (Deng et al., 2020; Huang et al., 2019; Li et al., 2018; Park et al., 2021; Sun et al., 2019; Wang et al., 2018; Xiao et al., 2019; Xu et al., 2020) [14,41,56,58,68,70,79,92], whereas there is much less information on soil microscopic fungi (Šimonovičová et al., 2019) [93]. The abundance and the diversity of a bacterial community vary along an Sb contamination gradient. The microbial populations (10^2.7^ cells/g) in the soils with Sb concentrations ranged from 12,433 to 21,400 mg/kg (Park et al., 2021) [68], and they decreased significantly in comparison to the microbial number in the bulk soil (10^8^ cells/g soil) (Raynaud and Nunan, 2014) [94]. The alpha diversity indices, including operational taxonomic units (OTUs), Simpson, Chao1, ACE, beta diversity, and microbial population size charactering microbial richness and evenness, decreased with the aggravation of Sb pollution (Park et al., 2021; Xiao et al., 2019) [68,70]. ACE, Chao1, Shannon, and Simpson indices decreased as the soil depth increased, which was probably due to a connection with the bioavailable Sb that increased with depth (Huang et al., 2019) [56].

*Proteobacteria*, *Actinobacteriota*, *Chloroflexi*, and *Acidobacteria* were the dominant bacteria phyla in the soils that were affected by the mining activities (Deng et al., 2020; Huang et al., 2019; Park et al., 2021; Wang et al., 2018; Xu et al., 2020) [41,56,58,68,92]. These dominant phyla are able to adapt to a harsh environment due to nitrogen fixation, phosphorus dissolution, and Sb and As oxidation (Xiao et al., 2019) [70]. Sb(III)-oxidizing bacteria at a genera level were found, including *Cupriavidus*, *Bacillus*, *Arthrobacter*, *Ensifer*, *Comamonas*, *Variovorax*, *Acinetobacter*, *Pseudomonas*, and *Stenotrophomonas* (Park et al., 2021) [68]. Novel hyper Sb-oxidizing bacteria (*Cupriavidus*, *Moraxella*, and *Bacillus*) were isolated from the contaminated mine soils and were found to be good candidates for Sb remediation in heavily polluted sites (Li et al., 2018) [14]. The effects of horizontal and vertical distribution patterns of Sb on microbial community compositions have also been analyzed (Park et al., 2021; Huang et al., 2019; Xu et al., 2020) [56,68,92]. The abundance of the Sb-oxidizing bacteria, such as *Pseudomonas*, *Bacillus*, and *Cupriavidus,* was not impacted by the horizontal or the vertical Sb gradients (Park et al., 2021) [68]. Other studies have found that different responses to Sb contamination at different soil depths were observed among the different components of the microbial communities in an Sb smelting plant (Xu et al., 2020) [92]. *Chloroflexi* increased with an increase of depth, while *Actinobacteria* and *Proteobacteria* decreased with depth (Huang et al., 2019) [56].

##### Factors Controlling Microbial Community Compositions

Regression analysis (Xiao et al., 2019) [70], canonical correspondence analysis (Deng et al., 2020) [58], redundancy analysis (Huang et al., 2019; Park et al., 2021; Wang et al., 2018) [41,56,68], random forest (Sun et al., 2019; Xiao et al., 2019) [70,79], and co-occurrence networks (Xiao et al., 2019) [70] were employed in order to investigate the microbial community composition in response to soil physicochemical properties, Sb contamination (concentration and speciation), and heavy metal(loid)s that occur as combined pollutants in Sb-contaminated sites.

Soil physicochemical properties, such as pH, oxidation-reduction potential (Eh), total sulfur (TS), proportion of carbon/nitrogen (C/N), total organic carbon (TOC), sulfate, and iron (Fe), as well as contamination fractions of Sb, As, Pb, etc., are the environmental drivers of the distribution of the dominant genera (Deng et al., 2020; Xiao et al., 2019; Xu et al., 2020) [58,70,92]. Nutrients promote the basic metabolism of bacteria (Wang et al., 2017) [95], while heavy metals reduce or even eliminate the metabolic capacity of carbon and nitrogen (Li et al., 2015) [96]. The importance of various factors influencing the microbial communities varied with the soil types. The pH had a greater impact on the microbial communities in the agricultural soils, whereas the Sb and the As extractable components had a greater effect on the microbial communities in the grassland and the forest soils (Sun et al., 2019) [79]. The pH can directly or indirectly change soil physical and chemical properties, which may drive the observed changes in the microbial communities at different concentrations of H^+^ (Boer et al., 2012) [97]. Many bacteria taxa have nearly neutral intra-cellular pH levels and can continue to grow within a narrow pH scope, i.e., three to four pH units (Rousk et al., 2010) [98].

As a majority of Sb is in the form of stable phase(s) with low bioavailability, Sb has a marginal effect on the microbial community composition (Park et al., 2021) [68]. However, soil Sb contamination was significant in shaping the microbial community structure compared to the soil physicochemical properties (Xiao et al., 2019) [70]. In particular, the bioavailable Sb had greater influences on the microbial community than the total Sb (Huang et al., 2019; Xiao et al., 2019) [56,70]. The bioavailable Sb, which was measured by DGT, was the dominant factor influencing the composition and the diversity of the bacteria (Wang et al., 2018) [41]. The reducible fraction of Sb had a positive correlation with *Chloroflexi* and *Rokubacteria* but was negatively correlated with *Proteobacteria* and *Actinobacteria* (Huang et al., 2019) [56]. The Sb that was distributed in the lithosphere is mainly associated with As and other metals. Co-contaminants may play a more important role in the control of the microbial diversity in Sb-contaminated sites. Occasionally, the impact of other heavy metals at high levels in the exchangeable and the organic-bound fractions on bacterial community structure was more remarkable than that of the Sb. The exchangeable fraction of As (Xiao et al., 2019) [70] and the extractable Pb (Park et al., 2021) [68] were the main factors affecting the microbial community compositions (Table 2).

## 4. Conclusions and Perspectives

The mining area, as the top of Sb resources in the industrial production chain, inevitably caused pollution to the surrounding bodies of water, soil, and plants. In this study, we summarized the spatial distribution, the pollution level, the chemical speciation, the influencing factors, and the ecological effects of Sb on the soils in mining/smelting areas. A total of 72% of the soils had Sb concentrations that were above the maximum permissible limit that is recommended by WHO. High soil Sb concentrations were detected in the developed countries (Poland, Italy, Spain, Portugal, New Zealand, and Australia) and some of the developing countries (China, Algeria, and Slovakia). The distribution of the study areas varies between the continents, which could have biased the results. The comparison of Sb chemical species in the soils of the different regions is difficult because of the differences in the extraction methods, the reagents, and/or the operating conditions that were used for chemical speciation analysis. The studies on the bioaccessibility of Sb in mining/smelting-affected soils are very limited. The clay content and the EC were positively correlated with Sb concentrations in the soil, and the Fe/Al hydrous oxide content in clay might be the dominant factor controlling the Sb accumulation in soils. The non-carcinogenic risks that the soil Sb pollution posed to the public in most of the regions were within the acceptable levels. However, more attention should be paid to the health risks near the heavily polluted mining areas. *Cynodon dactylon*, *Amaranthus paniculatus*, *Boehmeria*, *Equisetum arvense L*., and *Pteris Vittata* showed good Sb accumulation properties. The enrichment ability of Sb by crops was very low. However, the potential non-carcinogenic risks to the local inhabitants from Sb through the crops in the mining areas deserve more attention. The total Sb and bioavailable fraction may greatly affect the microbial community compositions, besides the soil variables (pH, Eh, TS, C/N, TOC, and sulfate) and the heavy metal(loid)s (As, Pb, and Fe).

There is still a lack of systematic research on the transport and the transformation over long distances and the biogeochemical cycle of Sb in the environment media of mining areas. There are numerous soil- and water-related national and international toxicity guidelines, legislations, and clean-up target values for Sb in order to evaluate the associated risks and the health hazards. It is necessary to establish universally acceptable regulatory guidelines and standards considering the Sb bioavailability, geography, socio-cultural aspects, and regulations, which would greatly improve the reliability and the robustness of the current techniques that are used for environmental management and the remediation of contaminated soils. Food ingestion represents the main source of human exposure to environmental pollutants. The accumulation of Sb in food crops could pose health risks to the residents in the area surrounding the mine. There is a need to identify effective methods, such as screening the crop species according to their potential to absorb or exclude Sb, or establishing legal thresholds for Sb in the edible parts of crops, to reduce the exposure risk through the consumption of locally produced food crops. The control of Sb pollution in mining areas should not only remediate the polluted area but also effectively control the pollution source by scientific management and mature technology. The lack of scientific understanding of the contribution and mechanism of natural and anthropogenic sources to Sb accumulation in soil restricts the scientificity of the existing prevention and control strategies of soil Sb pollution to a certain extent.

## Figures and Tables

**Figure 1 ijerph-20-00242-f001:**
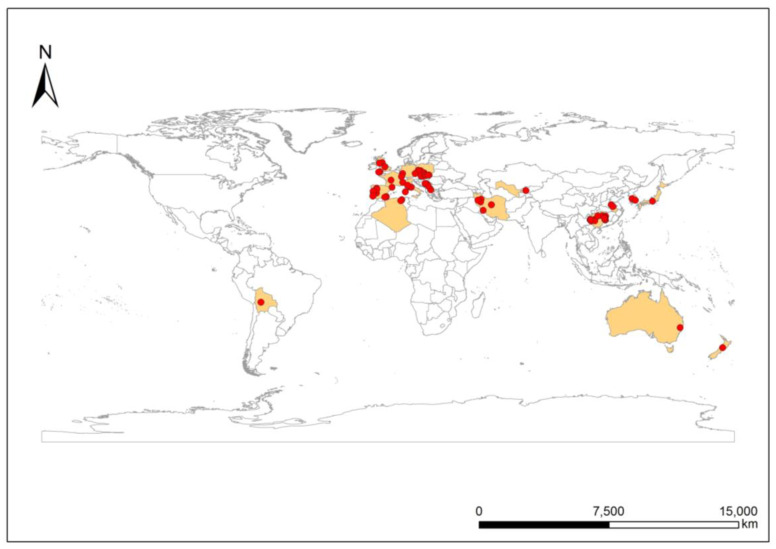
Geographical location of Sb studies related to mining/smelting activities retrieved from the scientific literature.

**Figure 2 ijerph-20-00242-f002:**
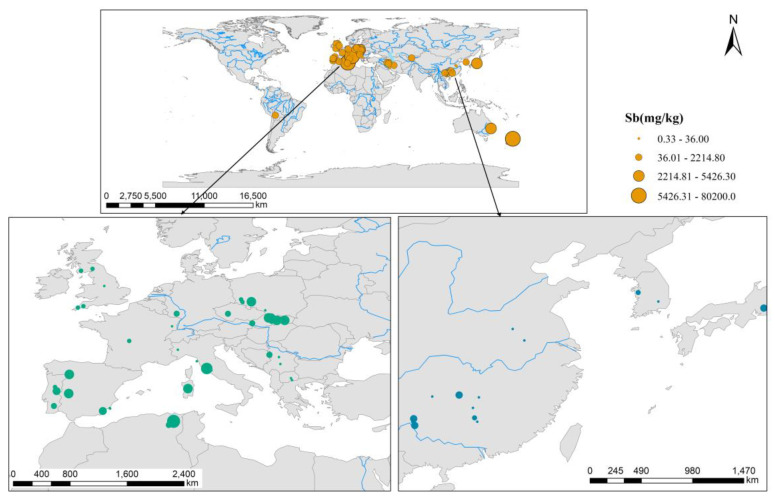
Spatial distribution of Sb concentrations in soils from different regions based on existing data. Specific regions from China and the European countries are enlarged. Changing the color of the orange dots into green and blue is just for clarity.

**Figure 3 ijerph-20-00242-f003:**
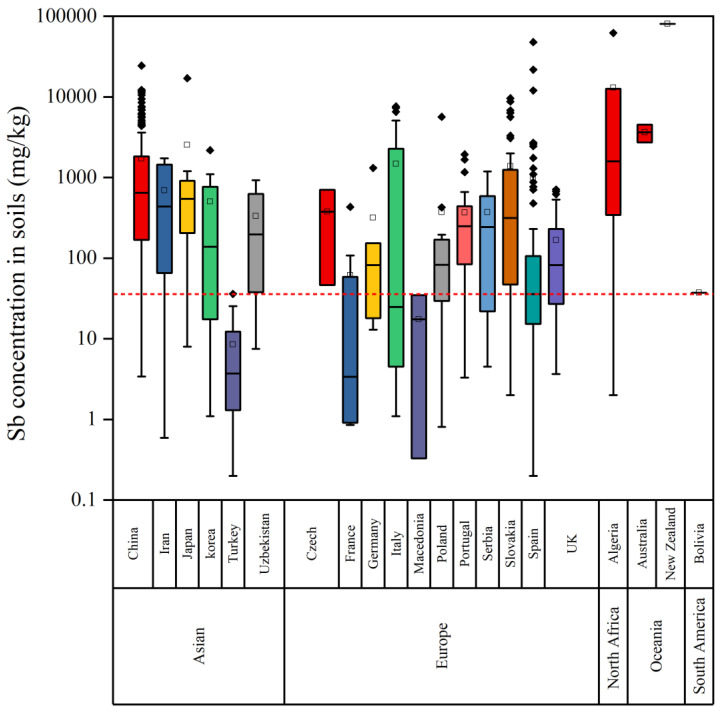
Global comparison between the average concentrations of Sb reported in soils related to mining/smelting activities. The red dotted line represents the maximum permissible limit (36 mg/kg) recommended by WHO.

**Figure 4 ijerph-20-00242-f004:**
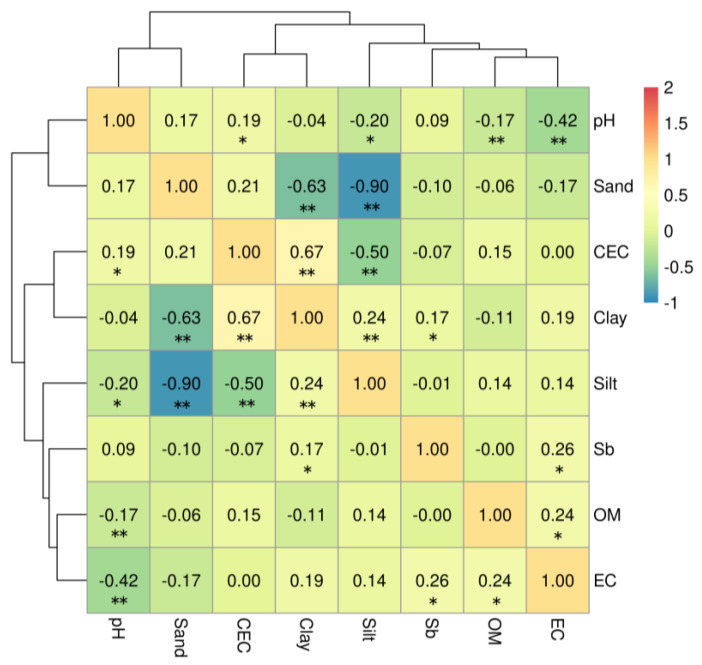
Heat map of correlation coefficients and clusters between Sb concentration and soil physicochemical properties. * represents *p* < 0.05 and ** represents *p* < 0.01.

**Figure 5 ijerph-20-00242-f005:**
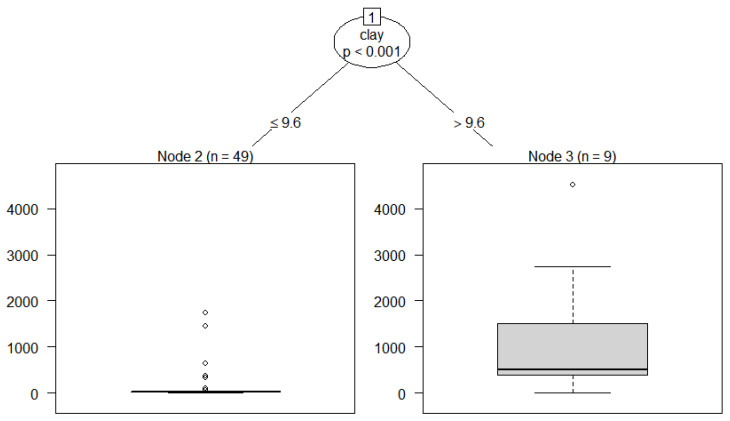
CIT analyses for Sb in soils (*n*: the number of samples; unit: mg/kg).

**Figure 6 ijerph-20-00242-f006:**
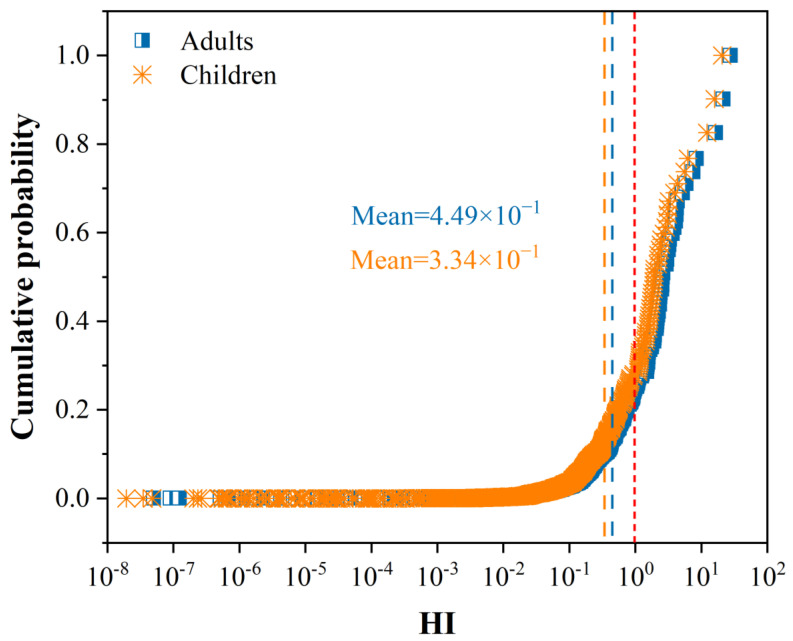
Cumulative probabilities of different population groups’ non-carcinogenic risk of Sb exposure in soils.

**Figure 7 ijerph-20-00242-f007:**
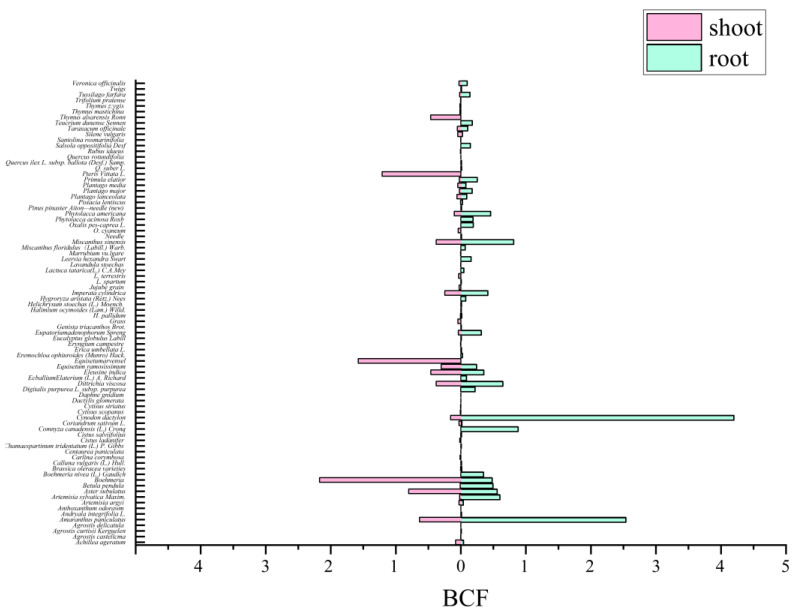
The ratio between the concentration of Sb in the root/shoot of plant and that present in the soil.

**Figure 8 ijerph-20-00242-f008:**
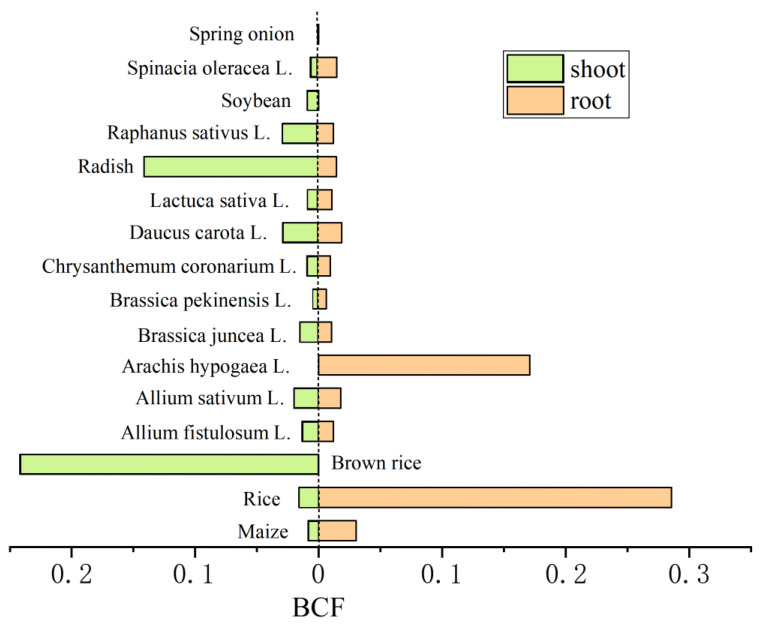
The ratio between the concentration of Sb in the root/shoot of crops and that present in the soil.

**Table 1 ijerph-20-00242-t001:** Summary of Sb speciation in soils related to mining/smelting activities.

Fraction of Sb in Soils						Procedure and Reagent	Reference
**Exchangable**	**Reducible**	**Oxizable**	**Residual**			BCR	Carvalho et al., 2012 [55]
0.05–1.38%	0.15–4.07%	0.11–1.34%	84.55–108.4%				
exchangable	specifically-sorbed surface-bound						Xiao et al., 2019 [70]
13.64%	9.17%						
exchangable	reducible	oxizable	residual				Protano and Nannoni, 2018 [57]
0.48–5%	1.82–7%	3.04–40%	60.47–95.95%				
complexing-reducing medium	acidic medium	oxidising medium	olsen extraction			acidic medium (1 g of solid in 25 mL 0.5 M HCl); the classical complexing-reducing medium containing citrateand dithionite, which is an oxidising medium containing H_2_O_2_, HNO_3_, and a 0.5 M NaHCO_3_ (Olsen extraction)	Pérez-Sirvent et al., 2011 [61]
25.6–81.01%	1.76–9.89%	0.28–4.51%	0–1.63%				
water-soluble						water: soil = 1000 mL: 100 g, shaken for 24 h	Murciego et al., 2007 [47]
0.74–2.45%							
water-soluble						water: soil = 10:1, shaken for 24 h	Casado et al., 2007 [46]
0.02–0.27%							
water-soluble						water: soil = 10 mL: 1 g, shaken for 2 h	Flynn et al., 2003 [60]
0–41.64%							
water-soluble						water: soil = 20 mL: 2 g, shaken for 2.5 h	Li et al., 2020 [53]
0.01–0.76%							
water-soluble							Wei et al., 2015 [63]
0.36–2.7%							
water-soluble						water: soil = 1:1 (*v/w*), shaken for 24 h	Pérez-Sirvent 2012 [61]
0.004–0.3%							
water-soluble	Sb(III)	Sb(V)				water: soil = 16 mL: 1.6 g, shaken for 24 h	Okkenhaug et al., 2011 [40]
0.6–10.2%	0.001–0.04%	0.59–10.15%					
water-soluble	extractable					water: soil = 100 mL: 50 g, shaken for 24 h; acetic acid: soil = 200 mL 0.43 mol acetic acid: 5 g, shaken for 16 h	Baroni et al., 2000 [44]
0.01–0.34%	0.03–8.62%						
water soluble	extractable	carbonatic	Fe and Mn oxides	sulphidic/organic	residual	Tessier method	He 2007 [38]
0.09–2.5%	0.31–2.11%	0.19–2.76%	0.52–3.39%	0.91–4.59%	88.2–97.92%		
	extractable	carbonatic	Fe–Mn oxides	sulphidic/organic	residual		Deng et al., 2020 [58]
	0.82–2.88%	2.88–9.74%	11.76–22.51%	3.87–6.11%	56.84–81.4%		
	extractable	carbonatic	Fe and Mn oxides	sulphidic/organic	residual		Ning et al., 2015 [52]
	0.15–2.48%	0.35–10.57%	0.79–3.89%	0.99–8.05%	84.55–95.98%		
soluble						1 M NH_4_NO_3_	Gál 2006 [65]
0.74%							
ionically bound	strongly absorbed	carbonates, Mn, Fe, Al oxides	co-precipitated with amorphous Fe oxides	co-precipitated with crystalline Fe oxides	co-precipitated with silicates	0.05 M (NH_4_)_2_SO_4_; 1 M NaH_2_PO_4_; 1 M HCl; 0.2 M NH_4_-oxalate; 1 M NaHCO_3_; 10 M HF; 16 N HNO_3_ +30% H_2_O_2_; HNO_3_:HCl:H_2_O = 1:3:4	Tan et al., 2018 [64]
1.56–1.67%	14.6–24.8%	9.02–11.1%	2.27–3.88%	14.7–27.3%	0.97–1.7%		
sulphidic/ organic	residual						
1.4–3.9%	27.1–52.6%						
CaCl_2_	Na_2_HPO_4_	HCl	EDTA	tartaric acid	citric acid		
0.6–0.86%	3.05–5.1%	2.46–6.95%	5.99–17.7%	9.01–14.97%	16.25–23.72%		
NH_4_NO_3_	CaCl_2_					1M NH_4_NO_3_; 0.01 M CaCl_2_	Lewińska et al., 2018 [64]
0.02–0.66%	0.05–1.12%						
water	NH_4_NO_3_	acetic acid	EDTA			water; 1 M NH_4_NO_3_; 0.05 M EDTA; 0.43 M acetic acid	Vaculík et al., 2013 [62]
0.03–10.11%	0.004–1.13%	0.11–9.33%	0.12–10.07%				
EDTA						0.05 M EDTA	Mariet et al., 2016 [66]
2.6–7.9%							
water	CaCl_2_	NH_4_NO_3_	DTPA	Na_2_HPO_4_		water; 0.01 M CaCl_2_; 1 M NH_4_NO_3_; 0.005 M DTPA; 0.1 M Na_2_HPO_4_	Ettler et al., 2007 [59]
1.55–1.75%	0.94–1.12%	0.86–0.89%	0.67–1.47%	2.2–9.11%			
SBET	PEBT Gastric	PEBT Intestinal				SBET; PEBT	Li et al., 2014 [39]
0.13–5.67%	0.49–13.25%	0.44–4.57%					
DGT (μg/L)						DGT	Wang et al., 2018 [41]
11.55–876.6							

**Table 2 ijerph-20-00242-t002:** Microbial community in soils from mining-contaminated environments.

Site	Basic Information	Dominant Microbial Community	Precentage/%	Note	Environmental Drivers of Microbial Community Structure		Reference
**Chungcheongnam-do, Republic of Korea**	Sb refinery	*Proteobacteria*	29.6	*Gammaproteobacteria, Alphaproteobacteria*	Pb might play a role in the differences in microbial community compositions; Sb content cannot explain the differences in microbial community composition.		Park et al., 2021 [68]
		*Acidobacteria*	23.1				
		*Chloroflexi*	11.8				
		*Actinobacteria*	8.8				
		*Arthrobacter*	0.04	Sb-oxidizing bacteria			
		*Bacillus*	0.38				
		*Ensifer*	0.01				
		*Comamonas*	0.05				
		*Cupriavidus*	0.14				
		*Variovorax*	0.03				
		*Acinetobacter*	0.06				
		*Pseudomonas*	0.86				
		*Stenotrophomonas*	0.02				
		*p_WPS_2*		Sb > 10,000 mg/kg			
		*o_Subgroup2*					
		*o_KF_JG30_C25*					
		*o_Subgroup13*					
		*f_Acidiferrobacteraceae*					
		*g_Sulfurifustis*					
		*o_Acidiferrobacterales*					
		*g_Granulicella*					
		*o_Betaproteobacteriales*		Sb > 200 mg/kg			
		*o_Acidobacteriales*					
		*p_Verrucomicrobia*					
Xiaohe tailing dump, Guizhou, Southwest China	Sb tailing dump	*Devosia*		N fixing		As_srp_, Sb_tot_, and Sb(V)-C were the main influence factors. The relative importance of Sb_exe_, Sb_srp_, and As_exe_ >10%.	Xiao et al., 2019 [70]
		*Cellvibrio*	3.14		C/N, Assrp,		
		*Lysobacter*	4.26		C/N, Sulfate		
		*Cohnella*					
		*Flavobacterium*	2.97	P solubilizing	C/N, TOC, Assrp, Sb(V)		
		*Paenibacillus*		Sb and As oxidation			
		*Bacillus*	5.39				
		*Pseudomonas*	6.25				
		*Thiobacillus*					
		*Agrobacterium*					
		*Corynebacterium*					
		*Methylotenera*					
		*Mycoplana*					
		*Paenisporosarcina*					
		*Pedobacter*					
		*Sphingobium*	5.76				
		*Yonghaparkia*					
		*Janthinobacterium*	2.94				
		*Sphingomonas*	3.08				
Dushan County, Guizhou, Southwest China	Banpo antimony mine and Xiaohe antimony mine mmelter	*Chloroflexi*			Positively associated with As_rec_, As_tot_, and Sb_rec_	The direct impact of As contamination fractions on bacterial community structure was greater than Sb, while the direct impact of Sb contamination fractions on bacterial function was more remarkable than As.	Huang et al., 2019 [56]
		*Rokubacteria*					
		*Proteobacteria*			Negatively associated with As_rec_, As_tot_, and Sb_rec_		
		*Actinobacteria*					
Lengshuijiang City, Hunan Province	Yanshan Sb mine smelter	*Acidobacteria*			Sb_tot_, As_tot_, pH, and Eh are more important based on the RF model; different components of the microbial communities responded differently to Sb and As contamination at different soil depths.		Xu et al., 2020 [92]
		*Chloroflexi*					
		*Proteobacteria*					
		*Thaumarchaeota*					
Lengshuijiang City, Hunan Province	Xikuangshan Sb mine	*Proteobacteria*			pH, Sb_DGT_, and As_DGT_ emerged as the most important factors.		Wang et al., 2018 [41]
		*Acidobacteria*					
		*Chloroflexi*					
		*Bacteroidetes*					
		*Actinobacteria*					
		*Gemmatimonadetes*					
		*Cyanobacteria*					
Lengshuijiang City, Hunan Province	Xikuangshan Sb mine	*Proteobacteria*		36.2–83.2			Deng et al., 2020 [58]
		*Acidobacteria*		36.2–83.2			
		*Bacteroidetes*		2.4–12.9			
		*Actinobacteria*		1.3–12.3			
		*Planctomycetes*		0.3–11.3			
		*Sideroxydans*				Sb_tot_, Sb_pavail_	
		*Luteolibacter*				Sb_avail_	
		*Povalibacter*				Sb_tot_, Sb_pavail_, Sb_avail_	
		*Lacibacterium*				Sb_avail_	
		*Gemmatimonas*				Sb_avail_	
		*Pirellula*				Sb_avail_	
		*Gp*				Sb_tot_, Sb_avail_	
		*Hydrogenophaga*				Sb_tot_, Sb_avail_	
		*Sphingomonas*				Sb_tot_, Sb_avail_	
		*Arthrobacter*				Sb_tot_, Sb_avail_	
		*Noviherbaspirillum*				Sb_tot_, Sb_avail_	
		*Escherichia/Shigella*				Sb_tot_, Sb_pavail_, Sb_avail_	
		*Arthrobacter*				bioremediation potential for Sb control	
		*Escherichia/Shigella*				bioremediation potential for Sb control	
Banská Štiavnica-Šobov, Zemianske Kostoľany, Smolník, Slovinky, Poproč, Slovakia	Old environmental loads from mining activities	*Penicillium chrysogenum var. chrysogenum*				The highest biodiversity of microfungal community was recorded in the extreme acidic environment, followed by the neutral, the ultra-acidic, and the very strong acidic ones.	Šimonovičová et al., 2019 [93]
		*Aspergillus niger*					
		*Neosartorya fischeri*					
		*Bionectria ochroleuca*					
		*Lewia infectoria*					
		*Phoma macrostoma*					
		*Phlebia acerina*					
Xiaohe Yelian smelting factory near the Dushan Sb mining area	Qinglong Sb mining area	*Proteobacteria*		12–50		The primary controlling factor of community richness was Sb_tot_, which explained 7% of the variation, followed by Fe(II) (6.7%) and pH (5%).	Sun et al., 2019 [79]
		*Alphaproteobacteria*		12.7	Proteobacteria		
		*Betaproteobacteria*		7.6			
		*Gammaproteobacteria*		5			
		*Deltaproteobacteria*		3.5			
		*Acidobacteria*		7.9–68			
		*Actinobacteria*					
		*Chloroflexi*					
		*Planctomycetes*					
		*Bacteroidetes,*					
		*Firmicutes*					
		*Geobacter*			As(V)-reducing bacteria		
		*Pseudomonas*			As(V)-reducing or As(III)-oxidizing bacteria		
		*Geobacter*			core microbiota in soils contaminated by As and Sb		
		*Pseudomonas*					
		*Janthinobacterium*			Sb-rich habitats		
		*Bradyrhizobium*			contain known nitrogen fixing members		
		*Rhodoplanes*					
		*Burkholderia*					
		*Clostridium*					
		*Corynebacterium*					

Note: Mtot—total concentration, Msrp—specifically-sorbed surface-bound fraction, Mexe—easily exchangeable fraction, Mrec—reducible fraction, MDGT—bioavailable fraction using DGT, Mpavail—poorly available fraction (carbonate + Fe–Mn oxides + organic), Mavail—exchangeable fraction.

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
