# Peer review of "Evaluation of Pollution Level, Spatial Distribution, and Ecological Effects of Antimony in Soils of Mining Areas: A Review"

_ijerph, 2022, doi:10.3390/ijerph20010242_

Round 1

Reviewer 1 Report

The paper is providing a literature review of the pollution level, spatial distribution and Environmental effect of Antimony in soils of mining areas.

I recommend changing the title, instead of environmental effect, use another term.

Engish is very difficult to understand. 

The paper is addressing the problem of Antimony in soils in the mining areas, but in the text, it was connected with the Sb deposits all over the world. It is important to mention Sb accumulation in the soil can be the result of other heavy metal extraction, such as Pb, Zn, so connecting the Sb pollution with Sb global reserves, lines 72 and 73, is not correct. 

What is Sb deposit- Line78

Introduction:  You mention Sb as metalloid and metal at the same sentence.

"Heavy metas may adsorb onto aeolian particles can be transported over long distances"-What does it mean?

"Data sources and extraction" -The term extraction is used in metallurgy, try to find some other expression

Line 86, please, explain

Line 138- Please explain

For division into regions the authors take levels of economic development, emission control technologies and distribution of Sb resources, line 156- Please be specific- what is considered as Sb resources?

Lines 186-190, not clear at all. What is the purpose of that statement?

The level of contamination and Environmental risk in the soil is usually estimated by sequential extraction analysis- it was enough to describe the procedure once, and than make a table with results. 

On the other hand, the relationship between clay and Sb behavior is analysed, as well as with microorganisms. 

The conclusion should resume all findings and give the answer on the research question. Please, rewrite it.

Author Response

I recommend changing the title, instead of environmental effect, use another term.

Response/action: Thank you very much for your suggestions. We changed the title as: Evaluation of Pollution Level, Spatial Distribution, and Ecological Effects of Antimony in Soils of Mining Areas: A Review

English is very difficult to understand.

Response/action: The English language was edited by Elsevier Language Editing Express Services before submission.

We have checked and modified the details and grammatical mistake in this revision.

Current line: 29, 30, 60, 100, 422.

The paper is addressing the problem of Antimony in soils in the mining areas, but in the text, it was connected with the Sb deposits all over the world. It is important to mention Sb accumulation in the soil can be the result of other heavy metal extraction, such as Pb, Zn, so connecting the Sb pollution with Sb global reserves, lines 72 and 73, is not correct.

Response/action: Thank you for pointing this out, and we have deleted the sentences of line 75, 76.

What is Sb deposit- Line78

Response/action: What we stress here is that China has the largest Sb reserves in the world. Elevated levels of Sb in soil on the one hand reflect an Sb-rich natural background. We have reorganized the expression.

Current line: 81-82

Introduction: You mention Sb as metalloid and metal at the same sentence.

Response/action: Thank you for your reminding. We have modified the contradictory statements. Current line: 39

"Heavy metal may adsorb onto aeolian particles can be transported over long distances"-What does it mean?

Response/action: According to Grousset et al. (1995), high enrichment factor of some heavy metals, such as As, Pb, Cd, and Sb were found and atmospheric input of anthropogenic origin can be clearly detected in the trap samples. This evidence demonstrated that Sb can transport over long distances by adsorbing aeolian particles.

reference

Grousset, F.E., Quetel, C.R., Thomas, B., Donard, O.F.X., Lambert, C.E., Guillard, F.,  Monaco, A., 1995. Anthropogenic vs. lithogenic origins of trace elements (As, Cd, Pb, Rb, Sb, SC, Sn, Zn) in water column particles: northwestern Mediterranean Sea. Marine Chemistry 48, 291-310. 

"Data sources and extraction" -The term extraction is used in metallurgy, try to find some other expression

Response/action: Thank you very much for your suggestions. We have changed extraction into collection.

Current line: 124

Line 86, please, explain

Response/action: Thanks for the advice. We have added some information here.

Current line: 91-93

Line 138- Please explain

Response/action: Thanks for the advice. We have added some explanation here.

Current line: 145-147

For division into regions the authors take levels of economic development, emission control technologies and distribution of Sb resources, line 156- Please be specific- what is considered as Sb resources?

Response/action: Thank you for pointing this out. The “Sb resources” herein referred to the typical Sb deposit. The regional division entirely based on the Zhu et al., 2020. We have revised the statement according to the reference.

Current line: 171-172

Lines 186-190, not clear at all. What is the purpose of that statement?

Response/action: In these sentences, we want to list the mining activities in detail. However, we agree with the reviewer’s assessment. Accordingly, throughout the manuscript, we have deleted those sentences for concise structure.

Current line: 201-206

The level of contamination and Environmental risk in the soil is usually estimated by sequential extraction analysis- it was enough to describe the procedure once, and than make a table with results.

Response/action: Extraction procedure, reagent and experimental conditions applied to obtain soil Sb speciation fraction are different for each study. To provide a straight comparison we summarize it in Table 1.

On the other hand, the relationship between clay and Sb behavior is analyzed, as well as with microorganisms.

Response/action: Thank you for your advice. We have added some information of the relationship between clay and the microorganisms.

Current line: 317-319

The conclusion should resume all findings and give the answer on the research question. Please, rewrite it.

Response/action: We consider your suggestion valuable. We have rewritten and summarized the conclusion and Perspectives.

Current line: 513-529

Reviewer 2 Report

This article is good to follow, with the proper format. The authors compiled Sb polluted soil data from 91 articles published and analysed pollution global distribution, evaluated Sb geochemical speciation extracting procedures, assessed soil Sb non-carcinogenic risk, and presented Sb plant uptake and microbial diversity and control factors. The authors also suggested some further research advice in soil Sb pollution.

For this paper to be published, the authors should provide a detailed and plausible explanation for the findings drawn by the authors and correct some expression and grammar mistakes.

Validity of the findings

1. The authors suggested that high Sb values mainly occurred in developed and developing countries and soil Sb concentrations decreased in the order of Oceania > North Africa > Europe > Asia > South America. Obviously, the authors did not distinguish between soils in mining, milling and smelting areas, and soils affected by these mining activities. For example, one soil sample with the highest Sb content (80200 mg/kg) in New Zealand is from the smelter (123500 mg/kg Sb in slag) in cited reference [86]. Soil samples with high Sb content in Australia is the contaminated mine sites capped by tailings (reference [88]). Soil samples from reference [35] include those from smelter and tailing sites and soils affected by the smelting and milling. The authors also did not provide referenced papers to support that soils have the lowest Sb content in Bolivia, Turkey, and Macedonia. Clearly, soils from different sites have different Sb contents, physical and chemical features, Sb speciation, bioavailability, and microbial diversity. The author should clearly state the different soil samples in the relevant parts of the article.

2. In the section of chemical speciation and bioavailability the authors compiled single extraction and sequential extraction procedures, DGT and EXAFS for Sb chemical speciation, bioaccessibility. However, soil Sb bioavailability is not well constrained by comparing different extraction procedures. The idea that the extractant selectivity lacks specificity and can only provide speciations in an operational sense is the common knowledge. Therefore, the authors do not add new knowledge to Sb soil availability study. 3. The authors found that clay content and electrical conductivity (EC) were positively correlated with Sb concentrations in soil (p < 0.05). However, the explanation is questionable. The explanation that the presence of Sb salts in soil lacks evidence and contradicts previous view that soil Sb has low bioavailability.

4. Many factors control microbial community and compositions of Sb polluted soils, the authors should list the order of the factors in different soils.

Additional comments

I list some expression mistakes in the following paragraphs, the authors should check the manuscript carefully to make it well expressed.

Line 16-17: Soil Sb concentrations reported in most countries significantly exceeded the maximum permissible limit recommended by WHO. It is the sampling bias responsible for the conclusion because cited articles are related to soils contaminated by Sb mining. 

Line 17-18: Soil Sb concentrations decreased in the order of Oceania > North Africa > Europe > Asia > South America. It did not provide useful information. The authors should use soil average Sb concentration in these regions. 

Lines 24: open-sky should be open-pit.  

Line 38: The expression as an “industrial monosodium glutamate” is not clear, I suggest delete it. 

Line 43: The expression “The State Council issued” is not correct. It should be “Ministry of Ecology and Environment of the People’s Republic of China released”. 

Line 43-44: The expression “on further strengthening heavy metal pollution prevention and control of views” is not clear. 

Line 45: I don’t understand what “particularly defend to” means. 

Line 50: What does "mass species” mean? 

Line 263-264: It was also reported that soil Sb in stibnite in Japan was mainly in the state of Sb(V) based on XANES analysis is incorrectly cited. “in stibnite” should be deleted in the sentence.

Line 286-287: The main components of clay are silicon dioxide, magnesium oxide, calcium carbonate, and aluminum oxide is incorrect. Clay in soil is mainly composed of phyllosilicate and iron oxides minerals.

Author Response

  1. The authors suggested that high Sb values mainly occurred in developed and developing countries and soil Sb concentrations decreased in the order of Oceania > North Africa > Europe > Asia > South America. Obviously, the authors did not distinguish between soils in mining, milling and smelting areas, and soils affected by these mining activities. For example, one soil sample with the highest Sb content (80200 mg/kg) in New Zealand is from the smelter (123500 mg/kg Sb in slag) in cited reference [86]. Soil samples with high Sb content in Australia is the contaminated mine sites capped by tailings (reference [88]). Soil samples from reference [35] include those from smelter and tailing sites and soils affected by the smelting and milling. The authors also did not provide referenced papers to support that soils have the lowest Sb content in Bolivia, Turkey, and Macedonia. Clearly, soils from different sites have different Sb contents, physical and chemical features, Sb speciation, bioavailability, and microbial diversity. The author should clearly state the different soil samples in the relevant parts of the article.

Response/action: Thank you for your advice. We have added this information of different activities related to mining in the manuscript in this revision.

Current line: 181, 469.

  1. In the section of chemical speciation and bioavailability the authors compiled single extraction and sequential extraction procedures, DGT and EXAFS for Sb chemical speciation, bioaccessibility. However, soil Sb bioavailability is not well constrained by comparing different extraction procedures. The idea that the extractant selectivity lacks specificity and can only provide speciations in an operational sense is the common knowledge. Therefore, the authors do not add new knowledge to Sb soil availability study.

Response/action: This study is a summary of existing research results on the bioavailability of Sb in soil from mining/smelting area and made a comparison of the distribution of Sb speciation fraction in soil in different research areas. We pointed out the problems of bioavailable Sb concentration in current research.

  1. The authors found that clay content and electrical conductivity (EC) were positively correlated with Sb concentrations in soil (p < 0.05). However, the explanation is questionable. The explanation that the presence of Sb salts in soil lacks evidence and contradicts previous view that soil Sb has low bioavailability.

Response/action: We consider your suggestion valuable. We have revised the inaccurate expression.

Current line: 299-301

  1. Many factors control microbial community and compositions of Sb polluted soils, the authors should list the order of the factors in different soils.

Response/action: In this study, we summarized the main factors controlling microbial community compositions from existing studies in Table 2. To identify qualitatively the relative importance of each factor, some model such as random forest model can be used. Based on limited data, this study can not identified qualitatively of main factors controlling microbial community compositions. The order of the factors was present based on the original result in some of studies in Table 2, such as Sun et al., 2019.

Additional comments

I list some expression mistakes in the following paragraphs, the authors should check the manuscript carefully to make it well expressed.

Line 16-17: Soil Sb concentrations reported in most countries significantly exceeded the maximum permissible limit recommended by WHO. It is the sampling bias responsible for the conclusion because cited articles are related to soils contaminated by Sb mining.

Response/action: Thank you for pointing this out. We have added more description for soil samplers. The revised text reads as follows on “Sb concentrations of polluted soil from mining areas”

Current line: 16

Line 17-18: Soil Sb concentrations decreased in the order of Oceania > North Africa > Europe > Asia > South America. It did not provide useful information. The authors should use soil average Sb concentration in these regions.

Response/action: We consider your suggestion valuable. We have added the average value of each region.

Current line: 18-20

Lines 24: open-sky should be open-pit.

Response/action: Thank you for pointing this out. We have changed open-sky into open-pit.

Current line: 26

Line 38: The expression as an “industrial monosodium glutamate” is not clear, I suggest delete it.

Response/action: Thank you for your advice. We have deleted the vaguely expression.

Current line: 39

Line 43: The expression “The State Council issued” is not correct. It should be “Ministry of Ecology and Environment of the People’s Republic of China released”.

Response/action: Thank you for pointing this out, and we have changed the expression according to your suggestion.

Current line: 44

Line 43-44: The expression “on further strengthening heavy metal pollution prevention and control of views” is not clear.

Response/action: Thank you for your advice. This is an opinion issued by the Ministry of Ecology and Environment of the People’s Republic of China. We have re-translated the title in this revision.

Current line: 45

Line 45: I don’t understand what “particularly defend to” means.

Response/action: Sb is one of the key focus of prevention and control. We have revised the unclear expression.

Current line: 46

Line 50: What does "mass species” mean?

Response/action: Sorry for the spelling mistake, we have changed mass into mosses.

Current line: 53

Line 263-264: It was also reported that soil Sb in stibnite in Japan was mainly in the state of Sb(V) based on XANES analysis is incorrectly cited. “in stibnite” should be deleted in the sentence.

Response/action: Thank you for pointing this out, and we have deleted it.

Current line: 282

Line 286-287: The main components of clay are silicon dioxide, magnesium oxide, calcium carbonate, and aluminum oxide is incorrect. Clay in soil is mainly composed of phyllosilicate and iron oxides minerals.

Response/action: As suggested by the reviewer, we have revised the description of clay in this revision.

Current line: 307

Round 2

Reviewer 1 Report

I agree with the modifications

Reviewer 2 Report

I agree the revised manuscript to be accepted.